# Relationship between Parents’ Physical Activity Level and the Motor Development Level and BMI of Their Children

**DOI:** 10.3390/ijerph19159145

**Published:** 2022-07-27

**Authors:** Jacqueline Paez, Juan Hurtado, Tomas Reyes, Rosita Abusleme, Patricio Arroyo, Cristian Oñate

**Affiliations:** 1Escuela Educación Física, Pontificia Universidad Católica de Valparaíso, Viña del Mar 252000, Chile; juan.hurtado@pucv.cl (J.H.); rosita.abusleme@pucv.cl (R.A.); 2Escuela Educación Física, Universidad Playa Ancha, Valparaíso 2340000, Chile; tomas.reyes@upla.cl; 3Escuela Educación Física, Universidad San Sebastián, Santiago 8320000, Chile; patricio.arroyo@uss.cl; 4Escuela de Kinesiologia, Universidad Católica de Temuco, Temuco 4780000, Chile; kinecristianjose@gmail.com

**Keywords:** parental attitude, physical activity, motor development, nutrition status

## Abstract

All the variables that arise in family dynamics can have significant effects on the lives of children concerning their nutritional status and motor development. The objective of this study was to relate the PAL of parents to the level of motor development and the BMI of their children. A total of 198 subjects participated, with the age of the students ranging between 8 and 10 years. To measure the BMI, the norms for the nutritional evaluation of children and teenagers from 5 to 19 years were used; to identify the motor behaviors, the TGMD-2 Test was used; and to identify the level of physical activity of parents, the International Physical Activity Questionnaire (IPAQ) was used. For the analysis, the independent samples t-test and the non-parametric Mann–Whitney U test (Wilcoxon) were used, and for the correlational analysis, Spearman’s rho test was applied. This study found no significant correlation between the activity level of parents and nutritional status variables (*p* = 0.162), or between the PAL variables of parents and the motor development of their children (*p* = 0.738). A parent’s level of physical activity does not have a direct relationship with the nutritional status or the motor development of their children.

## 1. Introduction

Physical activity (PA) is related to several distinct elements of health [1,2]. It is directly related to physical, cognitive, and psychosocial health [3,4], and to the time in front of a screen and the regulation of sleep duration [5]. PA is defined as a period in which a person focuses on light, moderate, or intense exercise, including activities from a daily walk to activities that raise the heart rate to high levels [5]. Staying physically active is important for preventing morbidities that lead to chronic diseases or death. Currently, low levels of physical activity in the global population represent a serious problem [6,7]. In general, low levels of physical activity are related to poor nutritional habits, obesity, and a sedentary lifestyle [4,5,6,8].

The World Health Organization (WHO) has indicated that children and young people between 5 and 17 years should spend at least 60 min a day in moderate or vigorous physical activity, as performing physical activity helps younger people to develop a healthy locomotor system and cardiovascular system, to learn to control the neuromuscular system (coordination and movement control), and to maintain a healthy body weight.

Children’s health behaviors develop within an ecological niche, with the family environment as a factor of great importance [9,10,11]. Factors such as access to the media, parental behavior, influence of siblings, and family habits can cause sedentary behavior in children [12,13]. This is explained by social cognitive theory: the regulation of behavior entails learning by observation, whereby children and teenagers model their behaviors according to the agents to which they are exposed daily [14]. Networks of strong social support increase an individual’s self-efficacy in overcoming barriers to being physically active [15,16]. If children observe that their mothers and fathers engage in regular physical activity and value it, they are likely to adopt the same values and behaviors [5,17,18]. These family dynamics are understood as emotional and physical activities along with communication [19] and the behaviors in parental practices such as gestures, changes in the tone of voice, and spontaneous expressions of affection [20]. All the variables that arise in family dynamics can have significant effects on the lives of children, including their nutritional status and level of physical activity [5,19]. The time that parents spend with their children, as well as what they do with this time, is significantly associated with the children’s body mass index [21], parental practices, parenting styles, and the promotion and limitation of habits [5] Whether they perform physical activity together during the early years of childhood is very important, as doing so is positively associated with the child’s physical activity [22].

The practice of physical activity provides different motor experiences, which, during the first years of childhood, are important for the development of motor skills.

These motor skills advance from simple patterns to more complex movements, where the basic motor skills (BMS), or fundamental movements, represent the basic structure on which motor responses of greater complexity are built [2,23,24,25,26,27]. Progress is made in the development and acquisition of mature patterns of handling, locomotor, and balance behaviors [28,29], which allow children to interact with the environment. BMS development is influenced by biological, psychosocial, and environmental factors [23,30,31,32]. Based on the hypothesis that the development of BMS is affected by their evolution, a high body mass index has negative effects on the psychosocial component of the child; this is a factor that negatively influences the development of BMS in preschoolers, whereby overweight or obese children present lower motor competence than expected for their age in locomotion, manipulation, and balance [27,33,34].

The relationship between motor skills and physical activity levels (PAL) is highly correlated [2,3,4,34,35]. BMS and PAL are bidirectional; not participating in physical activity can be caused by poor MS, and not participating in physical activity limits the opportunity to improve skills [3,36]. Similarly, there is a positive association between vigorous physical activity performed by parents and their children, whereby young people with parents who reach the recommended weekly frequency of vigorous physical activity are more likely to participate in vigorous physical activity and to increase the number of days per week that they perform these practices compared to those children whose parents perform physical activity less frequently [37]. Teenagers are more likely to practice physical activity if their parents are practitioners [38]. When one of the parents is physically active, the children have a model at home that likely motivates them to practice physical activity, and if both are physically active, the possibility of adopting physical activity as part of their routine increases [17,39]. The objective of this study was to elucidate the relationship between the level of physical activity of parents and the level of motor development and the BMI of their children.

## 2. Materials and Methods

### 2.1. Participants

The sample was non-probabilistic and convenient. A total of 198 subjects participated, including 99 parents (72 mothers and 27 fathers) and 99 students (52 girls and 47 boys) from three schools in the commune of Viña del Mar, Chile. The average age of the students ranged from 8 to 10 years old, with a mean of 103 ± 15 (SD) months.

### 2.2. Procedure

The study protocol was conducted considering the ethical principles for research involving human subjects proposed by the Declaration of Helsinki (World Medical Association) and the procedural and documentation suggestions of the Research Directorate at Pontificia Universidad Católica de Valparaíso through their Scientific Ethics and Bioethics Committee (BIOEPUCV-H158-9-12-2018). Authorization by the school authorities was requested, and then an informed consent form detailing the objectives and scope of the study was sent to the parents and/or legal guardians to authorize the participation of their children.

### 2.3. Instruments

#### 2.3.1. Nutritional Status

To determine the nutritional status of the participants (underweight, normal weight, overweight, or obese) based on their body composition (BMI), the norms for the nutritional evaluation of children and adolescents from 5 to 19 years of age were used [40] (Chilean Ministry of Health). For this, a portable height rod (Bodymeter Seca 206) (SECA, Viña del Mar, Chile) and a digital scale (Scale plus Body Fat Monitor UM-028, TANITA) (TANITA, Viña del Mar, Chile) were used, and the ratio between weight and height, expressed in kg/mts^2^, was then calculated.

#### 2.3.2. Motor Development

To identify motor behaviors, the TGMD-2 Test—one of the most widely employed tests to assess motor competence in children—was used [3]. The purpose of this test is to determine the motor development of children between 3 years and 10 years and 11 months of age by categorizing motor behaviors into seven categories: very poor, poor, below average, average, above average, superior, and very superior. These categories are determined according to gender and age (in months), considering the evolution of the motor development of children. Gross motor skills are assessed and grouped into two subtests: locomotor skills (running, galloping, hopping on one foot, two-foot horizontal jumping, running over an obstacle, and lateral movement) and manipulative skills (receiving, bouncing, rolling, throwing, and hitting a ball with a bat or the foot). This is done to obtain three ratings: one for locomotor development, one for manipulative development, and one for gross motor development in general. Each gross motor skill includes three to four behavioral components presented as performance criteria, where a score of 1 is given if the task is performed correctly and a score of 0 is given otherwise. After the test is applied and the scores of the two attempts per test for both subtests are added, the obtained scores are analyzed using a conversion table according to the age in months of the children, giving a standard score as a result that determines the gross motor quotient, which is then rated, using the seven categories previously mentioned, based on its value: >130 = very superior; 121–130 = superior; 111–120 = above average; 90–110 = average; 80–89 = below average; 70–79 = poor; <70 = very poor.

#### 2.3.3. Physical Activity Level

To determine the physical activity level of the parents, the International Physical Activity Questionnaire [41]—which has a correlation coefficient of 0.65—was used. This questionnaire provides information on the estimated energy expenditure in 24 h, on time spent walking, and on moderate-intensity, vigorous-intensity, and sedentary activities. The short version of the questionnaire (9 items) was used. It assesses three specific characteristics of physical activity: intensity (low, moderate, or vigorous), frequency (measured in days per week), and duration (time per day). After calculating the Physical Activity Index, whose value is equal to the product of the intensity (in METs), the frequency, and the duration of the activity, the participants were classified into three categories based on the following conditions: Low (little to no physical activity; does not meet criteria for categories Moderate or High), Moderate (3 or more days of vigorous activity for at least 20 min per day, 5 or more days of moderate-intensity activity or walking of at least 30 min per day, or 5 or more days of any combination of walking, moderate-intensity activity, and vigorous-intensity activity achieving a minimum of at least 600 MET-min/week), and High (vigorous-intensity activity on at least 3 days accumulating at least 1500 MET-min/week, or 7 or more days of any combination of walking, moderate-intensity activity, or vigorous-intensity activity achieving a minimum of at least 3000 MET-min/week). The MET reference values are as follows: walking = 3.3 METs; moderate PA = 4.0 METs; vigorous PA = 8.0 METs.

### 2.4. Data Analysis Technique

The IBM SPSS Statistics 25 software (New York, NY, USA) was used for data analysis. The Kolmogorov–Smirnov test was used to determine the data distribution (*n* > 50), and then the independent samples t-test and the non-parametric Mann–Whitney U test (Wilcoxon) were applied to corroborate the heterogeneity of the samples and to determine the statistical significance at a 95% confidence level (*p* < 0.05). For the correlational analysis, Spearman’s rho (ordinal variables) was used to determine the existence of an association between the qualitative variables and to corroborate the statistical significance at a 95% confidence level (*p* < 0.05).

## 3. Results

Table 1 shows the characterization of the samples by gender and total group for the following variables: age, weight, height, BMI, nutritional status, motor development level of the children, and physical activity level of the parents. For numerical variables, their mean and standard deviation are shown. For categorical variables, their frequency and percentage are shown. Of the total sample, 47.5% had an overweight, obese, or severely obese nutritional status, with similar values between females and males (48.1% and 46.8%, respectively). Only 18.2% of the parents and legal guardians had a vigorous physical activity level, while the rest of the sample (81.8%) had a low or moderate level. Regarding motor development level, it was observed that 78.8% of the total sample had a very poor, poor, or average level, with similar values between females and males (78.8% and 78.7%, respectively). Only three students (one female and two males) had an above-average or very superior level. For all characterization variables, there was no significant difference between females and males.

Table 2 shows the relationship between the physical activity levels of the parents and the nutritional status of the students. Regarding the parents, 100% of those whose children had failed to thrive, 85.2% of those whose children were normal weight, 77.8% of those whose children were overweight, 70% of those whose children were obese, and 100% of those whose children were severely obese were revealed to have a low or moderate physical activity level. In comparison, minor percentages of the children of the 18 parents who revealed having a vigorous physical activity level fell in the normal weight, overweight, and obese categories. No significant correlation was identified (*p*-value = 0.162, with *p*-value <0.05 considered significant) between the physical activity level and the nutritional status variables.

Table 3 shows the relationship between the physical activity level of the parents or legal guardians and the motor development level of their children. Regarding the parents, 79.3% of those whose children had a very poor or poor motor development level, 80% of those whose children had a below-average motor development level, 94.4% of those whose children had an average level, 50% of those whose children had an above-average motor development level, and 100% of those whose children had a superior or very superior motor development level revealed having a low or moderate physical activity level. The majority of parents who revealed having a vigorous physical activity level had children with a very poor or poor motor development level. No significant correlation was identified (*p*-value = 0.738, with *p*-value < 0.05 considered significant) between the physical activity level of the parents and the motor development level variables.

## 4. Discussion

Parents are essential agents when it comes to encouraging, promoting, and transmitting learning related to the acquisition of healthy habits in their children. However, the results of this study do not show a significant correlation between the level of physical activity of parents and the nutritional status of their children. On the other hand, the results of this study suggest that those parents who have a low or moderate level of physical activity have more children who are overweight (38.9%), obese (38.9%), or even morbidly obese (100%). These results coincide with another study that investigated parental support for physical activity in 6-year-old children, which showed that 58% of parents declared that they did not perform physical activity during the day, and indicated that 90% of them did not meet weekly physical activity recommendations [42]. As mentioned above, one of the risk factors that influences the weight control of children is the practice of physical activity carried out by the person responsible for them—the father, mother, or legal guardian. In addition, the influence of the family is a factor that must be considered in the design of prevention strategies against childhood obesity [43].

Another study found a significant association between the behavior of fathers and mothers and their children’s behavior in terms of performing sedentary activities, such as watching TV, videos, or DVDs. Similarly, the study pointed out that parents are a factor that significantly influences children’s regular practice of physical activity. These results indicate that parents’ behavior determines, to some extent, the nutritional status of their children [44]. Another study indicated that 95.2% of the mothers participating in the study did not relate the effect of physical activity to the bodyweight of their children. The underestimation of this variable could explain the relationship between the low or moderate level of physical activity of parents and the nutritional status (such as being overweight or obese) of their children indicated in this study [45].

Regarding the intensity of the practice of physical activity, overweight and obese children have parents who practice non-federated recreational sports, influencing the intensity of the practice in the nutritional status of their children [46].

Similarly, another study highlighted that the support and influence of parents in the practice of physical activity of their children have a positive influence, reducing the average weekly time in front of screens, increasing the weekly time of physical activity, increasing the values of Vo2 max, and resulting in a lower BMI score. These antecedents could explain the relationship between the nutritional status of being overweight or obese in children and the low or moderate physical activity of parents. Regarding the relationship between the level of activity of the father, mother, or legal guardian and the level of general motor development of the students, the results of this study suggest that it is not possible to establish a statistically significant correlation, but that it is possible to suggest that parents with a low or moderate physical activity level have children with very poor, below-average, or average motor development scores on tests of motor development [47]. In this regard, other results agree that parents who are physically active and have a healthy lifestyle positively affect their children, both in their development of motor skills and their nutritional status [11]. At the same time, it is also possible to find agreement between the results of this study and other findings [48], which, when researching the relationship between gender, family context, and extracurricular physical activity and motor coordination of boys and girls aged 8 to 9 years, pointed out that there are no significant differences in the relationship between the level of parents’ physical activity and the scores on tests of locomotion and control of objects.

In addition, [49] identified a significant and sustained effect on the levels of physical activity in a sample of fathers and their daughters through the implementation of physical activity programs based on games, recreational activities, sports skills, aerobic physical activity, and muscle strengthening; both fathers and their daughters had an increase in steps per day. The results of this study show the influence that parents have on the level of physical activity and motor development of their children. This means that the lower the amount of physical activity of the parents, the lower the score of their children on motor development tests, and vice versa.

On the other hand, [50] pointed out that parents who do not perform enough physical activity during the month have children with poor performances in tests of basic motor skills, agreeing with the findings of this study, where the majority of children were in below-average categories of motor development and most parents had low to moderate PAL.

## 5. Conclusions

Promoting the practice of physical activity in the family unit is essential for encouraging the acquisition of healthy habits in children. In this sense, parents are an essential model for healthy behaviors and, in turn, have a high influence on the healthy behaviors of their children. The results obtained in this study show that parents with a low or moderate level of physical activity have children who are overweight or obese, and who have poor or very poor motor development. Despite the above, a trend can only be suggested in the relationship between the level of physical activity of parents and the nutritional status and motor development of their children, as no significant differences were obtained between the PAL of parents and the BMI and level of motor development of their children. In conclusion, it is necessary to collect more information about this relationship and request that those groups responsible for generating policies and interventions related to the acquisition of healthy life habits in children consider parental influence in achieving a substantial and sustained impact.

## Figures and Tables

**Table 1 ijerph-19-09145-t001:** Characterization by gender and total group of the children.

Characterization Variables	Females (*n* = 52)	Males (*n* = 47)	Total (*n* = 99)
Gender of children (%)	52.5	47.5	100.0
Age of children (mean ± *SD*) ^1^	101.0 ± 15.0	105.1 ± 14.9	103.0 ± 15.0
Weight of children (mean ± *SD*) ^1^	31.5 ± 7.1	31.9 ± 8.1	31.7 ± 7.5
Height of children (mean ± *SD*) ^2^	130.6 ± 9.6	131.5 ± 9.4	131.0 ± 9.5
Body mass index of children (mean ± *SD*) ^1^	18.2 ± 2.5	18.2 ± 2.8	18.2 ± 2.7
**Nutritional status of children ^1^**			
Failure to thrive *n* (%)	1(1.9)	2 (4.3)	3 (3.0)
Eutrophy *n* (%)	26 (50.0)	23 (48.9)	49 (49.5)
Overweight *n* (%)	20 (38.5)	16 (34.0)	36 (36.4)
Obese *n* (%)	4 (7.7)	6 (12.8)	10 (10.1)
Severely obese *n* (%)	1 (1.9)	0 (0.0)	1 (1.0)
**Motor development of children ^1^**			
Very poor *n* (%)	6 (11.5)	12 (25.5)	18 (18.2)
Poor *n* (%)	22 (42.3)	18 (38.3)	40 (40.4)
Below average *n* (%)	13 (25.0)	7 (14.9)	20 (20.2)
Average *n* (%)	10 (19.2)	8 (17.0)	18 (18.2)
Above average *n* (%)	1 (1.9)	1(2.1)	2 (2.0)
Superior *n* (%)	0 (0.0)	0 (0.0)	0 (0.0)
Very superior *n* (%)	0 (0.0)	1 (2.1)	1 (1.0)
**Physical activity level of parents ^1^**			
Low *n* (%)	22 (42.3)	21 (44.7)	43 (43.4)
Moderate *n* (%)	20 (38.5)	18 (38.3)	38 (38.4)
Vigorous *n* (%)	10 (19.2)	8 (17.0)	18 (18.2)

^1^ Mann–Whitney test; ^2^ independent samples *t*-test; *SD*: standard deviation.

**Table 2 ijerph-19-09145-t002:** Physical activity levels concerning nutritional status.

Physical Activity Level	Deficit Weight *n* = 3 (%)	Healthy Weight *n* = 49 (%)	Overweight *n* = 36 (%)	Obese *n* = 10 (%)	Severely Obese *n* = 1 (%)	*p* ^1^
Low	2 (66.7)	23 (46.9)	14 (38.9)	4 (40.0)	0 (0.0)	0.162 (r = 0.142)
Moderate	1 (33.3)	19 (38.8)	14 (38.9)	3 (30.0)	1 (100.0)	
Vigorous	0 (0.0)	7 (14.3)	8 (22.2)	3 (30.0)	0 (0.0)	
Total	100%	100%	100%	100%	100%	

^1^ Spearman’s rho; r: correlation coefficient.

**Table 3 ijerph-19-09145-t003:** Relationship between physical activity levels and motor development.

Physical Activity Level	Very Poor and Poor (*n* = 58) *n* (%)	Below Average (*n* = 20) *n* (%)	Average (*n* = 18) *n* (%)	Above Average (*n* = 2) *n* (%)	Superior and Very Superior (*n* = 1) *n* (%)	*p* ^1^
Low	25 (43.1)	10 (50.0)	8 (44.4)	0 (0.0)	0 (0.0)	0.738 (r = 0.034)
Moderate	21 (36.2)	6 (30.0)	9 (50.0)	1 (50.0)	1 (100.0)	
Vigorous	12 (20.7)	4 (20.0)	1 (5.6)	1 (50.0)	0 (0.0)	
Total	100%	100%	100%	100%	100%	

^1^ Spearman’s rho; r: correlation coefficient.

## Data Availability

Data are available upon reasonable request submitted to the corresponding author.

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
