# Peer review of "Relationship between Parents’ Physical Activity Level and the Motor Development Level and BMI of Their Children"

_ijerph, 2022, doi:10.3390/ijerph19159145_

Round 1
Reviewer 1 Report
Review: Relationship between the parent’s physical activity level and the motor development level and BMI of their children.
I applaud the authors for their time and effort in highlighting the importance of parental involvement in a child's engagement in PA. Please consider these suggestions.
Line 15: ages of the students ranging between…
Line 28-36: I would ask the authors to consider rewording this first sentence for clarity and purpose. The addition of lower use of screens and sleep regulation appears to be an after thought or its meaning isn’t as clear. I assume you are referring to screen time use and lower sleep regulation. Physical activity is also classified in modes of light PA, moderate, etc. This reads as though PA only constitutes moderate to intense (vigorous) levels. PA must be considered from both the functional side as well as the cardiovascular side. You also state low levels of PA are linked with intake of alcoholic drinks…is this a concern for the 8–10-year-old age group? What distinct variables are linked with youth and low PA levels?
Line 39-41: I would argue that ones’ ability to locomote and the prerequisites to do so effectively and efficiently fall underneath the neuromuscular system and should be listed as one unit. Coordination and control are the foundations of basic locomotor skills.
Line 38: remove coma
Line 40: system as well as learn to control
Line 41: and maintain a healthy body weight.
Line 45: theory; the information
Line 47: [14]. The networks of strong…
Line 55: I would suggest distinguishing that type of time a parent spends with their child in this statement. We know that it is the type and or quality use of time. Time being physically active versus time spent watching tv etc.
Line 59: This paragraph seems out of place following the previous train of thought. Also, it is important to note that motor movement learned in infancy do happen quickly; however, learning fundamental motor skills is a speedy process which is not learned solely on the dynamics of their environment. FMS need to be taught and the correlation between proficiency in a child’s FMS is linked with BMI and coordination.
Line 56: [21]. Also, if they…
Line 67: component of the child; it is…
Line 72: bidirectional; not participating…
Line 73: remove coma after MS
Line 75: children, whereas young people…
Line 82-83: [17,39]. The objective in this study is to…
Line 105: I would argue that the term nutritional status and BMI are two distinctly different constructs. Nutritional status implies you have assessed variables related to diet intake. Tanita does provide body weight status which is different.
Line 135: [41], … 0.65, was used.
Line 178: this study reveals that 100%...
Line 272: children. It is possible… or In conclusion, it is possible…
Author Response
Reviewer
We have made the requested changes and welcome your feedback and suggestions for improvement.
It improves in writing, relevance of results among others.
Please see the attachment.

Reviewer 2 Report
Congratulations for your work. The research has some originality and interest. However I have some concerns and suggestion:
1- Rewrite conclusion and discussion. Conclusion it’s not support by results (at max you can mention that results suggest a trend or link despite the no significant difference) and discussion need improve the language, clarity and the missing or inadequate citation;
2- You need a better explain about the limitation of your study;
Your methodology have gaps and/or need mor clarification/adequate citation (Citation at line 114 it’s not the TGMD-2 test or validation for your country) (Where from the sample? Citation of IPAQ validation is for Iranian children?!?) (Why 99 parents? Parents are poor explain and characterized. Are two (mother and father)? Are only one? If is one who is their? You explain at line 80-82 that there is difference if is one of the parents or if are both!
3- In general paper you should verify:
i. Citations are not adequate to the text (e.g. “…Physical activity (PA) is related to several distinctive elements of health [1,2], it is directly related to physical, cognitive, and psychosocial health [3, 4] and with lower use of screens and sleep regulation [5].” Citation 1,2 don’t aimed or conclude any, direct benefits of PA and Health; Citation 3,4 don’t aimed or conclude or measured the effect of PA on cognitive and psychosocial health. Other citation should be more adequate, .i.e., citation 8 is about university students and you have paper about children. Citation at line 114 it’s not the TGMD-2 test or validation for your country! Miss citation (221, 235, 248,254,267).
4- Complement table2 and 3 with total % for nutritional status and motor development
5- Review text writing: Line 44 to 49! And others (…..line 210….237….245… 249… 258..)
At the discussion I suggest: The results failed!?! (line 209) the results show… (line 210) the results indicates?!? The results suggest…
Abbreviation NAL don’t have description before.
Author Response
Reviewer
We have made the requested changes and welcome your feedback and suggestions for improvement.
It improves in writing, relevance of results among others.
Please see the attachment.

This manuscript is a resubmission of an earlier submission. The following is a list of the peer review reports and author responses from that submission.